# First Identification, Recombinant Production, and Structural Characterization of a Putative Structural Protein from the Haseki Tick Virus Polyprotein

**DOI:** 10.3390/biom15121690

**Published:** 2025-12-03

**Authors:** Irina A. Osinkina, Alexey O. Yanshin, Egor O. Ukladov, Yury L. Ryzhykau, Alexander P. Agafonov, Anastasia V. Gladysheva

**Affiliations:** 1State Research Center of Virology and Biotechnology “Vector”, 630559 Kol’tsovo, Russia; osinkina_ia@vector.nsc.ru (I.A.O.); yanshin_ao@vector.nsc.ru (A.O.Y.); agafonov@vector.nsc.ru (A.P.A.); 2Physics Department, Novosibirsk State University, 630090 Novosibirsk, Russia; 3Scientific Educational Center Institute of Chemical Technology, Novosibirsk State University, 630090 Novosibirsk, Russia; e.ukladov@g.nsu.ru; 4SRF “SKIF”, 630559 Kol’tsovo, Russia; 5Research Center for Molecular Mechanisms of Aging and Age-Related Diseases, Moscow Institute of Physics and Technology, 141701 Dolgoprudny, Russia; rizhikov1@gmail.com; 6Frank Laboratory of Neutron Physics, Joint Institute for Nuclear Research, 141980 Dubna, Russia

**Keywords:** *Ixodid tick*, tick-borne infection, RNA viruses, novel virus, *Flaviviridae*, protein structure, small-angle X-ray scattering, AlphaFold

## Abstract

Haseki tick virus (HSTV) is a recently discovered virus detected in human serum following tick bites, yet its protein repertoire remains uncharacterized. In this study, we applied an integrative approach based first on membrane topology prediction, followed by AI-based structural prediction and experimental validation to annotate the structural part of the HSTV polyprotein. For the first time, we recombinantly expressed one of the putative HSTV structural protein (SP1) and determined its overall architecture using small-angle X-ray scattering (SAXS). Structural comparisons of the AI-predicted HSTV SP1 models revealed only a vague resemblance to the pestiviral Erns and Npro. The strong agreement between experimental SAXS data and the AI-predicted HSTV SP1 model supported the conclusion that HSTV SP1 adopts a distinct spatial architecture in solution, one that is not captured by existing pestiviral structures but is reliably represented by modern AI-based prediction. Our findings indicate that HSTV SP1 adopts a fold not previously observed among characterized members of the *Flaviviridae* family. This work establishes a methodological pipeline for characterizing highly divergent viral proteins and provides the first insights into HSTV SP1, a virus with emerging zoonotic potential. These results lay the foundation for future functional and structural studies, diagnostic development, and evolutionary analyses of atypical *Flaviviridae* family members.

## 1. Introduction

*Flaviviridae* is a family of viruses whose genomes consist of nonsegmented single-stranded positive-sense RNA. The genome of viruses in the *Flaviviridae* family typically contains a single open reading frame encoding a polyprotein that is post-translationally cleaved by viral and cellular proteases. The polyprotein encodes structural and non-structural proteins. Structural proteins are typically encoded at the N-terminus of the polyprotein and are part of the virion during the formation of new viruses. Non-structural proteins (including helicase, protease, and RNA-dependent RNA polymerase) are encoded at the C-terminus and participate in polyprotein processing and viral genome replication [1]. Many members of this family are capable of causing severe human diseases (e.g., dengue virus, Zika virus, tick-borne encephalitis virus) or economically significant animal diseases (e.g., bovine viral diarrhea virus, classical swine fever virus). The *Flaviviridae* family includes four genera: *Pestivirus*, *Hepacivirus*, *Orthoflavivirus*, and *Pegivirus* [1]. However, in recent years, the advent of high-throughput sequencing technologies has led to the discovery of a large number of novel viruses that may pose a threat to human health and necessitate further investigation and classification.

Among recently discovered viruses belonging to the *Flaviviridae* family that are potentially hazardous to human health and currently poorly characterized, particular attention has been drawn to novel viruses exhibiting atypical genome organization. Notably, this includes the Jingmenvirus group, whose members possess a segmented genome, as well as the proposed group of large-genome flaviviruses (LGF), which some researchers consider distinct [2]. The LGF group includes recently identified viruses such as *Bole tick virus 4* (BoTV4), *Trinbago virus*, and *Haseki tick virus* (HSTV). Trinbago virus was first detected in *Dermacentor reticulatus* and *Ixodes persulcatus* ticks in China [3]. BoTV4 has been detected in *Dermacentor reticulatus* ticks in China and Eastern Europe, as well as in *Rhipicephalus* spp. ticks in Kenya [4]. HSTV has been detected in *Ixodes persulcatus* ticks collected in Russia [5], the Mediterranean region, and Georgia, and has also been identified in human serum samples following tick bites in individuals presenting with clinical signs of illness [6]. The evidence of potential human infection with this virus, combined with its classification within the *Flaviviridae* family that includes epidemiologically significant pathogens, underscores the necessity for further HSTV investigation.

Traditional approaches to virus characterization and classification primarily rely on phylogenetic analysis of highly conserved proteins, particularly the RNA-dependent RNA polymerase (RdRp) [7]. However, in the case of HSTV, the amino acid sequence homology with other known viruses is less than 30%, rendering homology-based identification of proteins unfeasible. Nevertheless, previous studies employing a structural bioinformatics approach specifically modeling segments of the nonstructural region of the HSTV polyprotein and comparing the resulting models with experimentally determined spatial structures of known nonstructural viral proteins successfully enabled the annotation of HSTV nonstructural proteins [8]. Moreover, structural modeling and analysis of the putative glycoproteins of BoTV4, the closest known relative of HSTV, revealed substantial deviations from the canonical glycoprotein architectures characteristic of *Flaviviridae* family members [9]. To date, the BoTV4 genome, as well as those of other closely related viruses recently discovered within the same clade, remains largely unannotated (Appendix A).

Viral structural proteins play a crucial role in virion assembly, protection of the viral nucleic acid, and mediating interactions with target host cells. However, they are often encoded by non-conserved genomic regions that can differ substantially even among closely related viruses [10]. As a result, structural proteins remain unidentified or poorly characterized in many recently discovered members of the *Flaviviridae* family, such as HSTV [11]. This severely hampers the investigation of the molecular mechanisms underlying the HSTV life cycle and impedes the development of POC-tests and immunoprophylactic strategies.

In this study, we aimed to overcome the limitations of conventional sequence-based approaches in annotating the HSTV structural proteins by employing an integrative strategy that combines genomic bioinformatic analysis, deep-learning-based structural prediction, and experimental validation.

## 2. Materials and Methods

### 2.1. Generation and Analysis of the HSTV SP1 Structural Model

The complete genomic nucleotide sequence of HSTV was retrieved from NCBI GenBank (National Library of Medicine, Bethesda, MD, USA) (accession ID: MW808978); the corresponding polyprotein sequence is available under GenBank accession ID: UTQ11742.

Transmembrane domain prediction was performed using three independent web servers: PredictProtein (Technological University of Munich, Munich, Germany; https://predictprotein.org/ (accessed on 1 March 2025)) [12], CCTOP v1.1.0 (Protein Bioinformatics Research Group, Institute of Enzymology, RCNS, Budapest, Hungary; https://cctop.ttk.hu/ (accessed on 1 March 2025)) [13], and TMHMM 2.0 (Department of Health Technology, Kgs. Lyngby, Denmark; https://services.healthtech.dtu.dk/services/TMHMM-2.0/ (accessed on 1 March 2025)) [14]. Protein domain annotation was carried out by searching the Pfam and NCBI Conserved Domain Database (CDD) using HMMER v3.1b2 (University of California, Santa Cruz, CA, USA; https://www.ebi.ac.uk/Tools/hmmer/ (accessed on 1 March 2025)) [15]. Multiple sequence alignments of nucleotide and amino acid sequences were generated with Clustal Omega v1.2.2 (EMBL, Dublin, Ireland; https://www.ebi.ac.uk/jdispatcher/msa/clustalo (accessed on 1 March 2025)) [16].

Structural models of putative HSTV structural proteins and their complexes were predicted using AlphaFold3 v3.0.1 via the official AlphaFold Server (Google DeepMind, London, UK; https://alphafoldserver.com/ (accessed on 1 March 2025)) [17]. Structural homologs were identified by querying the Protein Data Bank (Research Collaboratory for Structural Bioinformatics Protein Data Bank (RCSB PDB), Piscataway, NJ, USA; https://www.rcsb.org/ (accessed on 1 March 2025)) [18] with the FoldSeek web service v10 (Seoul National University, Seoul, South Korea; https://search.foldseek.com/ (accessed on 1 March 2025)) [19]. Additional structural models of the HSTV SP1 protein were generated using RoseTTAFold (University of Washington, Seattle, WA, USA; https://robetta.bakerlab.org (accessed on 1 August 2025)) and Boltz (Massachusetts Institute of Technology, Boston, MA, USA; https://boltz.bio/ (accessed on 1 August 2025)).

Comparison of secondary structures and search for conserved sequence regions of the HSTV SP1 with Erns and Npro of the *Flaviviridae* family was visualized using the ESPript v.3.0 software (Institute of Protein Biology and Chemistry, Lyon, France, https://espript.ibcp.fr/ (accessed on 1 August 2025)) [20].

Tertiary structure models of HSTV proteins were visualized using Mol*Viewer v.4.10.0. (Protein Data Bank in Europe (PDBe), Hinxton, UK; Research Collaboratory for Structural Bioinformatics Protein Data Bank (RCSB PDB), Piscataway, NJ, USA; CEITEC—Central European Institute of Technology, Brno, Czech Republic; ELIXIR CZ, Praha, Czech Republic), https://molstar.org/ (accessed on 1 August 2025)) [21].

### 2.2. Design and Synthetic Assembly of DNA Constructs Encoding the HSTV SP1

The nucleotide sequence encoding the HSTV SP1 was codon-optimized for expression in a prokaryotic system using the Benchling (Benchling, San Francisco, CA, USA; https://benchling.com/ (accessed on 1 April 2025)).

Synthetic DNA constructs encoding HSTV SP1 were assembled by overlap extension PCR using partially complementary oligonucleotides with 20-nucleotide overlaps (Appendix A). PCR amplification was performed with Q5 High-Fidelity DNA Polymerase (NEB, Hitchin, UK). At each assembly step, amplicons were analyzed by electrophoresis in a 2% agarose gel and subsequently purified using the Cleanup Standard kit (Eurogen, Moscow, Russia). For long-term storage and downstream applications, the assembled DNA fragments were cloned into the high-copy-number blunt-end cloning vector pJET1.2/blunt (Thermo Fisher Scientific, Waltham, MA, USA), which enables efficient propagation of recombinant DNA in *E. coli*. Ligation products were transformed into chemically competent *E. coli* NebStable strain (NEB, Hitchin, UK) via heat shock and plated onto LB agar plates (AppliChem, Darmstadt, Germany) supplemented with ampicillin as a selective marker. The accuracy of DNA assembly and cloning at each step was verified by Sanger sequencing using the BigDye Terminator v3.1 Cycle Sequencing Kit (Thermo Fisher Scientific, Waltham, MA, USA) on an ABI 3500/3500xl Genetic Analyzer (Applied Biosystems, Waltham, MA, USA). Sequence data were processed and analyzed using the UGENE (UGENE, Novosibirsk, Russia). Plasmid DNA for experimental use was prepared by culturing transformed *E. coli* cells in liquid LB medium (AppliChem, Darmstadt, Germany) supplemented with ampicillin for 12 h at 37 °C, followed by purification with the Plasmid Miniprep kit (Eurogen, Moscow, Russia).

### 2.3. Generation of Recombinant HSTV SP1-Producing Strains

To construct expression vectors, the HSTV SP1 transgene was amplified by PCR using the previously generated plasmid pJET1.2/blunt-SP1 as a template. The resulting amplicons were cloned into the following expression vectors: pET200/D-TOPO (Thermo Fisher Scientific, Waltham, MA, USA), pEASY (TransGen Biotech, Beijing, China), and a custom-modified plasmid containing, within a single open reading frame, a 14×His affinity tag followed by a TEV-protease cleavage site, all under the control of a synthetic T7 bacteriophage promoter. Chemically competent *E. coli* cells of the following strains were transformed with the respective constructs using the heat-shock method: BL21(DE3), KRX (Thermo Fisher Scientific, Waltham, MA, USA), HyperStable(DE3) (Raissol Bio, Moscow, Russia), T7EvoPro (Raissol Bio, Moscow, Russia), and LysTech (Raissol Bio, Moscow, Russia). To optimize expression conditions, a series of small-scale production trials was performed, systematically varying cultivation parameters such as temperature, glucose concentration in the medium, and inducer concentration.

### 2.4. Expression, Extraction, and Purification of Recombinant HSTV SP1

Recombinant protein expression was carried out by cultivating the producer strain in Terrific Broth medium for 20 h at 20 °C and 180 rpm, with induction by 1 mM IPTG (Thermo Fisher Scientific, Waltham, MA, USA) at an optical density (OD600) of 0.6. Following cultivation, cells were harvested by centrifugation at 10,000 rpm for 20 min and stored at −60 °C. For protein extraction, the cell pellet was resuspended in 20 mM Tris-HCl buffer (pH 8.0) and lysed using a high-pressure homogenizer YC-NANO10 (Yocell Biotechnology, Qingdao, China). The resulting lysate was clarified by centrifugation at 10,000 rpm for 40 min.

Protein purification was performed in two sequential steps, ion-exchange chromatography (IEC) followed by size-exclusion chromatography (SEC), using an HBBio-Lab 100 Chromatography System (Hanbon Sci. & Tech., Huaian, China) with UV detection at 280 nm. IEC was carried out on a ProteomixPor15-Q anion-exchange resin (Sephax, Newark, DE, USA) using a linear pH gradient from 10.0 to 4.0. The column was equilibrated with five column volumes of 20 mM Tris-HCl (pH 10.0). A stable linear pH gradient was generated by gradually mixing the starting buffer (20 mM Tris base, pH 10.0) with the elution buffer (150 mM Tris base, pH 4.0) until 100% elution buffer was reached. Protein fractions collected at different pH values were analyzed by SDS-PAGE. Molecular weight estimation was performed using Precision Plus Protein All Blue Standards (Bio-Rad, Hercules, CA, USA). Fractions containing the target protein were dialyzed overnight (16 h) against 20 mM Tris-HCl, 150 mM NaCl (pH 8.0) and concentrated to ~10 mg/mL using 10 kDa MWCO centrifugal filter units (Jet BioFil, Guangzhou, China).

SEC was performed on a pre-packed Superdex 200 Increase 10/300 GL column (GE Healthcare, Stockholm, Sweden), equilibrated with three column volumes of 20 mM Tris-HCl, 150 mM NaCl (pH 8.0). A 500 µL aliquot of the concentrated sample was loaded onto the column and eluted at a flow rate of 0.4 mL/min. Eluted fractions were collected and analyzed by SDS-PAGE. Dynamic Light Scattering data were collected using a BeNano 180 Zeta Pro (Bettersize, Dandong, China).

### 2.5. Small-Angle X-Ray Scattering (SAXS) Data Collection and Analysis

SAXS measurements for HSTV SP1 were performed at the BL19U2 beamline of the Shanghai Synchrotron Radiation Facility (SSRF, Shanghai, China). The X-ray beam size on the stage was 0.33 mm (H) × 0.05 mm (V). A two-dimensional Pilatus3 2M detector (DECTRIS Ltd., Baden, Switzerland) was placed at a sample-to-detector distance of 2.7 m. The scattering vector magnitude range (q = (4π/λ)sinθ, where 2θ is the scattering angle and λ = 0.1033 nm is the wavelength) was 0.07–4.5 nm^−1^.

Scattering profiles were collected at protein concentrations of 12.31 and 3.12 mg/mL in 20 mM Tris-HCl, 150 mM NaCl (pH 8.0), with 20 frames of 1.0 s exposure each. The SAXS data were collected at room temperature. Buffer scattering was measured under identical conditions. For each measurement, the 20 frames were averaged separately for the buffer and for the protein solution. The final scattering curve was obtained by subtracting the averaged buffer scattering from the averaged protein scattering. Data were processed using BioXTAS RAW [22] and analyzed with the ATSAS v.4.0.1 suite (EMBL, Dublin, Ireland; https://www.embl-hamburg.de/biosaxs/ (accessed on 1 August 2025) [23].

The radius of gyration (*R*_g_) and forward scattering intensity I(0) were determined from Guinier analysis (valid for s*R*_g_ < 1.3; s*R*_g_ = 0.35–1.28 for the 3.12 mg/mL dataset and s*R*_g_ = 0.50–1.27 for the 12.31 mg/mL dataset). *R*_g_ was estimated from the Guinier approximation using PRIMUS. Molecular weight (MW) was estimated using a Bayesian approach as implemented in the Molecular Weight Wizard of PRIMUS (ATSAS 3.0). Pair-distance distribution functions P(r) were computed with GNOM with automatically determined s-ranges to obtain *R*_g_ (s = 0.014–0.328 1/Å and *R*_g_ = 25.2 Å for the 3.12 mg/mL dataset; s = 0.021–0.336 1/Å and *R*_g_ = 24.1 Å for the 12.31 mg/mL dataset) and the maximum particle dimension (*D*_max_) [24].

Low-resolution ab initio shape reconstructions were generated using DAMMIF [25], and multiple independent models were averaged and validated with DAMAVER/SUPCOMB [26,27]. The excluded volume was estimated from ab initio modeling results generated by DAMMIF. Theoretical scattering curves from AlphaFold3-predicted, RoseTTAFold-predicted, and Boltz-2-predicted HSTV SP1 models were calculated with CRYSOL [28] and compared to experimental data to assess structural consistency. The goodness of fit was assessed using the reduced, weighted χ^2^ statistic, as implemented in CRYSOL and DAMMIF.

## 3. Results

The positions of protein domains within the structural region of the HSTV polyprotein could not be reliably predicted by searching for conserved viral homologous domains in the Pfam and NCBI Conserved Domain Database (CDD). Although an HMMER-based hidden Markov model search revealed high sequence similarity to the polyprotein of Bole tick virus 4 (E-value = 0), it failed to resolve the domain architecture.

Therefore, we employed an alternative strategy to delineate protein domains in the structural region of the HSTV polyprotein by first predicting its membrane topology and subsequently modeling the resulting extramembrane and cytoplasmic parts using AlphaFold3.

### 3.1. Identification of a Putative Structural Protein 1 from the Haseki Tick Virus Polyprotein

#### 3.1.1. Membrane Topology

Within the putative structural region of the HSTV polyprotein (residues 1–1250), three transmembrane domains were identified at the following positions: 174–189 (16 a.a.), 652–671 (20 a.a.), and 832–851 (20 a.a.). Based on this topology, the extramembrane and cytoplasmic parts were delineated as follows:Extramembrane regions: residues 1–173 (173 a.a.) (soluble ectodomain of putative structural protein 1, SP1) and 672–831 (160 a.a.) (soluble ectodomain of putative structural protein 3, SP3);Cytoplasmic regions: residues 190–651 (462 a.a.) (soluble ectodomain of putative structural protein 2, SP2) and 852–1250 (399 a.a.) (soluble ectodomain of putative structural protein 4, SP4) (Figure 1; Appendix A).

Tertiary structural models were generated for each of these domains using AlphaFold3. The resulting pLDDT were as follows: SP1—0.87, SP2—0.41, SP3—0.39, and SP4—0.43. The high pLDDT value for SP1 indicates a well-ordered, confidently predicted fold, suggesting that this segment (residues 1–173) likely corresponds to a structured protein or domain within the HSTV structural polyprotein region. In contrast, the low pLDDT scores for SP2, SP3, and SP4 reflect high structural uncertainty, precluding reliable structural interpretation or functional inference for these regions based on current AlphaFold-based modeling.

#### 3.1.2. SP1 Tertiary Structure Models

The HSTV SP1 tertiary structure model (173 a.a.) comprises two distinct domains connected by a flexible, unstructured linker of 9 a.a. Domain 1 (residues 1–84) consists of 84 a.a. and adopts a mixed α/β architecture composed of six α-helices and two β-strands arranged in the following topology: β1–α1–α2–α3–β2–α4–α5–α6. Domain 2 (residues 94–149) spans 56 a.a. and is characterized by a predominantly β-structural fold, containing three short α-helices and six β-strands organized as β1–β2–α1–α2–β3–β4–β5–β6–α3. Notably, the β-strands in HSTV domain 2 form a canonical “Greek key” motif, a structural hallmark commonly observed in viral capsid and envelope proteins. The C-terminal tail (residues 150–173) is predicted to be intrinsically disordered and lacks stable secondary structure elements.

Structural similarity between HSTV SP1 and pestiviral Erns

A structural homology search using FoldSeek revealed the highest similarity between the HSTV SP1 model and the ectodomain of the Erns glycoprotein precursor from *Bovine viral diarrhea virus 1* (BVDV-1, PDB ID: 4DVK), with a TM-score of 0.69 and RMSD of 3.51 Å (Table 1 and Appendix A).

The Erns (an RNase T2-family member) is a virion surface glycoprotein unique to members of the *Pestivirus* genus. Comprising 227 amino acid residues, Erns plays a critical role in viral entry by binding to the host cell receptor ribosomal protein SA (RPSA), a cellular adhesion molecule that functions as a viral attachment factor. In addition to its structural role, Erns exhibits ribonuclease activity that contributes to immune evasion by suppressing type I interferon production in infected cells, thereby promoting viral infection [30]. The catalytic site of pestiviral Erns is formed by the conserved residues His32, His76, Glu77, Lys80, and His81 [29]. Upon structural superposition, these residues align closely with His15, His63, Glu64, Lys67, and His68 in our AlphaFold3-predicted model of HSTV SP1 (Figure 2a,b). The apparent shift in residue numbering arises from the absence of the N-terminal 15 a.a. in SP1, which are present in Erns. In the experimentally determined Erns structure, the catalytic residues are coordinated by a sulfate ion. Although AlphaFold3 does not model inorganic cofactors such as sulfate ions, the spatial arrangement of the corresponding side chains in the SP1 model closely recapitulates that observed in Erns, suggesting that SP1 may similarly accommodate a structurally analogous anion or cofactor in its active site.

In the HSTV SP1 structural model, four pairs of cysteine residues occupy spatial positions analogous to those in the Erns structure (PDB ID: 4DVK) and are predicted to form disulfide bonds: Cys28–Cys70, Cys57–Cys58, Cys95–Cys142, and Cys99–Cys125 (Figure 2c). In Erns, these disulfide bridges are critical for maintaining the protein’s tertiary architecture. Notably, the Cys28–Cys70 disulfide bond is highly conserved among RNase Rh/T2/S family members [31]. Furthermore, the SP1 model contains a vicinal disulfide bond between adjacent cysteine residues (Cys57–Cys58) previously described as unique to Erns among pestiviral proteins (Figure 2c) [31].

The surface electrostatic potential map of the HSTV SP1 structural model reveals an asymmetric charge distribution between the convex and concave surfaces of the protein. The inner surface, which lines the concave fold, exhibits a pronounced positive electrostatic potential, consistent with a potential role in RNA binding. In contrast, the outer convex surface displays a predominantly negative electrostatic potential (Figure 3). A similar asymmetric electrostatic potential distribution has been reported for the Erns structure [29].

Given the structural and sequence-level parallels between HSTV SP1 and pestiviral Erns, including the conserved catalytic residues, disulfide bond pattern, and overall fold, we hypothesized that the full-length functional counterpart of SP1 might correspond to an Erns-like protein of 227 amino acids. To test this, we generated an AlphaFold3 model of the N-terminal 227 residues of the HSTV polyprotein (residues 1–227), extending beyond the soluble SP1 ectodomain (residues 1–173) to encompass the predicted C-terminal membrane-associated region.

The AlphaFold3-predicted model of the N-terminal 227 a.a. of the HSTV polyprotein, in addition to the previously described putative N-terminal RNase-like domain, features a C-terminal amphipathic α-helix spanning 26 residues. This helix closely resembles the C-terminal α-helix of the Erns, which functions as a membrane anchor [32] (Figure 4).

Furthermore, the position of the transmembrane domain in the HSTV polyprotein (residues 174–189) aligns with the location of the transmembrane segment in Erns, reinforcing the structural and topological similarity between the two proteins.

We further investigated whether HSTV SP1, either in its 172-residue (Figure 4b) or 227-residue (Figure 4c) form, could form homodimers analogous to Erns. In pestiviruses, homodimerization of Erns is mediated by an intermolecular disulfide bond between Cys171 residues (Cys171–Cys171′), which is essential for its oligomeric state [33]. However, to date, no crystallographic or high-resolution structural data of full-length, disulfide-linked Erns homodimers are available; existing structures represent only the N-terminal ectodomain (residues 1–154), lacking the Cys171-containing C-terminal region. Consistent with this, no intermolecular interaction involving the equivalent residue (Cys172 in HSTV) was observed in the AlphaFold3-predicted models of SP1 homodimers, suggesting that SP1 may not dimerize via this conserved cysteine-mediated mechanism (Figure 4).

Structural similarity between HSTV SP1 and pestiviral Npro

A recent study has proposed a major revision of the *Flaviviridae* family taxonomy [34]. According to this proposal, the traditional Flaviviridae clade should be restructured into three distinct families: *Flaviviridae* (*sensu stricto*), *Pestiviridae*, and *Hepaciviridae*. The closest known relative of HSTV, BoTV4, is suggested to be assigned to the *Pestiviridae* family under this new scheme. Given this proposed reclassification, we considered whether the N-terminally encoded HSTV SP1 might be functionally analogous to pestiviral Npro.

Npro is a protein unique to pestiviral members. It is the first protein encoded in the pestiviral polyprotein, comprising 168 amino acid residues. Npro undergoes autocatalytic cleavage from the nascent polyprotein during viral genome translation in infected cells and functions as a key immune evasion factor by interacting with interferon regulatory factor 3 (IRF3), thereby triggering its proteasomal degradation [35].

Notably, despite a low TM-score of 0.29, indicating limited structural similarity, visual inspection of the SP1 model and the N-terminal protease (Npro) from *Classical swine fever virus* (CSFV, PDB ID: 4H9K) revealed a moderate topological resemblance in their domain organization (Appendix A). The AlphaFold-predicted model of HSTV SP1 (173 a.a.) comprises two domains that together adopt a conformation resembling a molluscan shell. The catalytic cysteine residue of CSFV Npro (Cys69) is spatially aligned with the homologous Cys70 in HSTV SP1 (Figure 5b). However, the SP1 structural model lacks the histidine residue (His49 in Npro) that, together with Cys69 and a zinc ion, forms a hydrogen-bonded catalytic dyad essential for proteolytic activity (Figure 5b). The nearest histidine residues in HSTV SP1 (His68, His63, and His55) are located within an α-helical and are structurally inaccessible for interaction with Cys70. Furthermore, analysis of the SP1 sequence for conserved motifs did not identify any significant homology to canonical Npro (Figure 5a and Appendix A).

### 3.2. Recombinant Production of a Putative Structural Protein 1 from the Haseki Tick Virus Polyprotein

Our structural modeling of the HSTV SP1 reveals features reminiscent of both Erns and Npro pestiviral proteins, yet these similarities are insufficient for definitive functional annotation of the identified domain. To unambiguously determine the biochemical activity of HSTV SP1 and to resolve its spatial structure experimentally, recombinant production of SP1 is essential.

#### 3.2.1. Production of HSTV SP1 to Assess Its Similarity to Npro

To test the hypothesis that the HSTV SP1 belongs to Npro, we generated a recombinant expression construct. The plasmid, designated T7-14×His-TEV-SP1-PS-mNeonGreen, contains a single open reading frame (ORF). This ORF encodes a 14×His affinity tag, followed by a TEV protease cleavage site, the full-length SP1 protein (173 amino acids, including a putative proteolytic cleavage site between residues 168 and 169), and the fluorescent reporter protein mNeonGreen (Figure 6a). In our construct, the SP1 (1–173) boundary was chosen to encompass the region homologous to the Npro cleavage site (positions 168–169 in pestiviruses), under the hypothesis that SP1 might function as a Npro-like protease. Using this construct, we generated two recombinant *E. coli* producer strains: BL21(DE3)/SP1-PS-mNeonGreen and KRX/SP1-PS-mNeonGreen, both expressing the chimeric protein 14×His-TEV-SP1-PS-mNeonGreen. Small-scale expression trials demonstrated that both strains produce the target soluble recombinant protein (~48 kDa) when cultured at 30, 25, or 20 °C, 180 rpm, for 24 h in the presence of 1 mM IPTG and/or 0.1% rhamnose (Figure 6b and Appendix A).

Npro is a zinc-dependent autoprotease. To assess whether SP1 possesses proteolytic activity, we produced and purified the recombinant chimeric protein 14×His-TEV-SP1-PS-mNeonGreen under various conditions, including the addition of 1 mM and 10 mM ZnSO_4_. In none of the tested conditions did we observe proteolytic cleavage of the chimeric protein into the expected fragments (14×His-TEV-SP1: ~23 kDa and mNeonGreen: ~25 kDa), indicating that SP1 lacks detectable autoproteolytic activity (Figure 6c and Appendix A).

Thus, under the tested conditions, we were unable to detect any proteolytic activity associated with HSTV SP1. This finding is consistent with our earlier structural analyses, which revealed no significant similarity between the Npro and the SP1 model.

#### 3.2.2. Production of HSTV SP1 for Structural Studies

Due to the challenges commonly associated with recombinant expression of viral proteins, including low solubility, proteolytic instability, and difficulties in tag removal, we engineered a series of expression constructs with varied affinity tags (6×-His tag, 14×-His tag), protease cleavage sites (for TEV protease, enterokinase protease, and Picornain 3C protease), and fusion architectures (pET-HSTV-SP1, pEASY-HSTV-SP1, and 14×His-Tev-SP1). This strategy aimed to identify optimal conditions for high-yield production, efficient purification, and generation of tag-free HSTV SP1 suitable for structural studies. Also, to facilitate successful structural studies, the predicted intrinsically disordered C-terminal region was removed from the HSTV SP1-encoding sequence, resulting in a final SP1 construct spanning 158 a.a. (Figure 7).

Recombinant producer strains were constructed in several *E. coli* expression strains, namely BL21(DE3), KRX, HyperStable, T7-EvoPro, and LysTech, that support the synthesis of the chimeric recombinant proteins (Appendix A). Cultivation of the BL21(DE3)/14×His-Tev-SP1, KRX/14×His-Tev-SP1, HyperStable/14×His-Tev-SP1, T7-EvoPro/14×His-Tev-SP1, and LysTech/14×His-Tev-SP1 recombinant producer strains resulted in detectable expression of the target recombinant protein (~20.5 kDa) under all tested conditions, as confirmed by SDS-PAGE (Figure 7a and Appendix A). However, in all strains carrying the 14×His-TEV-SP1 construct, the recombinant protein was exclusively produced as inclusion bodies, while no detectable expression of the target chimeric protein (~20.8 kDa) was observed in the BL21(DE3)/pEASY-HSTV-SP1, KRX/pEASY-HSTV-SP1, and HyperStable/pEASY-HSTV-SP1 strains (Figure 7b and Appendix A).

The KRX/pET-HSTV-SP1 strain proved most effective for producing chimeric recombinant HSTV SP1 (~26 kDa, 207 a.a.) in soluble form. Cultivation for 20–22 h at 20 °C and 180 rpm, with induction by 1 mM IPTG and 0.1% L-rhamnose, resulted in high-level expression of soluble chimeric recombinant HSTV SP1 localized in the bacterial periplasm (Figure 7c and Appendix A). The theoretical isoelectric point (pI) of the chimeric recombinant HSTV SP1 was calculated to be 6.39. At pH 8.0, the chimeric recombinant HSTV SP1 carries a net charge of −4.1.

During IMAC, the chimeric recombinant HSTV SP1 failed to bind to the resin. This is likely due to the spatial conformation of the expressed chimeric protein, which may occlude the His-tag and is not reliably predictable by AlphaFold3.

Consequently, purification of the chimeric recombinant HSTV SP1 was performed in two steps using ion-exchange and size-exclusion chromatography (SEC). In ion-exchange chromatography, the recombinant SP1 eluted at pH 6.4–6.2 (Appendix A), in close agreement with its theoretical pI. SEC yielded a single symmetric peak at 16.7 mL (Figure 8a), corresponding to the expected elution volume for a monomeric protein of ~26 kDa. Moreover, dynamic light scattering (DLS) analysis revealed a hydrodynamic diameter of 4.4 nm ± 0.1 nm and a main peak mass of 96% for the chimeric recombinant HSTV SP1 protein, further supporting its monomeric state in solution (Figure 8b). The final purified preparation exhibited >90% homogeneity, as confirmed by SDS-PAGE (Figure 8c and Appendix A), making it suitable for structural studies.

### 3.3. Structural Characterization of a Putative Structural Protein 1 from the Haseki Tick Virus Polyprotein by SAXS

To gain insights into the overall architecture and oligomeric state of HSTV SP1 in solution, we performed small-angle X-ray scattering (SAXS) experiments at the BL19U2 beamline (Shanghai Synchrotron Radiation Facility, Shanghai, China). Data were collected for HSTV SP1 samples at two concentrations (12.31 and 3.12 mg/mL) to evaluate concentration-dependent aggregation and ensure the reliability of structural parameters. SAXS is particularly well suited for validating computational structural models (e.g., those generated by AlphaFold3) and for assessing the conformational homogeneity and spatial structure of proteins under near-physiological conditions.

Experimental SAXS profiles enabled the determination of key structural parameters for the HSTV SP1. Guinier analysis yielded a radius of gyration (*R*_g_) of 2.5 nm, while the pair-distance distribution function (P(r)) indicated a maximum particle dimension (*D*_max_) of 8.0–8.4 nm. The molecular weight of HSTV SP1, estimated using Bayesian inference, ranged from 28.9 to 33.1 kDa, slightly higher than the theoretical value of ~27 kDa. This discrepancy varied with protein concentration and may be attributed to several factors, including hydration effects and the presence of a flexible N-terminal tail associated with a 6 × His tag (Table 2).

At the HSTV SP1 lowest concentration tested (3.12 mg/mL), SAXS data are consistent with a monodisperse particle, yielding a *R*_g_ = 2.5 ± 0.1 nm and a *D*_max_ = 8.0 nm. Guinier, Kratky, and P(r) plots collectively indicate a predominantly compact conformation with moderate flexibility and no signs of aggregation in the low-angle region. Notably, the absence of an upward curvature (“upturn”) at the high-s end of the Kratky plot further confirms the lack of higher-order oligomers or aggregates (Figure 9a and Appendix A).

At the HSTV SP1 highest concentration tested (12.31 mg/mL), the primary scattering curve, log I(s), shows no evidence of aggregation or low-angle artifacts. The Guinier plot at low s values is well fit by a straight line, further supporting the absence of interparticle interactions or oligomerization. However, the Kratky plot exhibits a slight deviation from the characteristic bell-shaped profile expected for a compact, globular protein, suggesting the presence of conformational flexibility or partial disorder in solution (Figure 9b and Appendix A).

The close agreement between the *R*_g_ values derived from the P(r) function and those obtained from Guinier analysis in both cases confirms the internal consistency of the SAXS data (Figure 9c).

The experimental SAXS curves were compared with theoretical scattering profiles generated from the Alphfold3, RoseTTAFold, and Boltz-2 predicted models by calculating the reduced chi-squared (χ^2^) statistic. In an ideal scenario, χ^2^ should equal 1, indicating perfect agreement between model and experimental data. Quantitative comparison of experimental SAXS profiles with theoretical scattering curves generated from different AI-predicted models revealed the following reduced χ^2^ values: 2.2 and 2.7 for the AlphaFold3-predicted SP1 structural model, 2.2 and 3.0 for the Boltz-2-predicted SP1 structural model, and 1.7 and 1.5 for the RoseTTAFold-predicted SP1 structural model (Table 3). Lower χ^2^ values for the RoseTTAFold prediction suggest a closer agreement with the experimental data under the tested conditions.

The HSTV SP1 structural model generated by RoseTTAFold (*D*_max_ (envelope) = 8.7) exhibits greater compaction of both the N- and C-terminal regions compared to the model produced by AlphaFold3 (*D*_max_ (envelope) = 11.1) (Figure 10, Table 3). In contrast, the Boltz-2 model displays even more pronounced compaction (*D*_max_ (envelope) = 6.0) (Figure 10, Table 3). In the Boltz-2-predicted HSTV SP1 structure, the N-terminal segment, containing the 6 × His affinity tag, is buried within the protein core, rendering it inaccessible for binding to IMAC resin (Appendix A). This structural occlusion likely explains the absence of specific binding of recombinant HSTV SP1 observed during IMAC and anti-6 × His Western blots (Appendix A).

Thus, the SAXS data validate the overall architecture of HSTV SP1, support its monomeric state in solution, and inform future functional studies and supporting structural efforts, including construct optimization for X-ray crystallography.

## 4. Discussion

HSTV represents an emerging member of the *Flaviviridae* family with documented detection in human serum following tick bites and association with clinical symptoms, raising concerns about its potential zoonotic risk [5]. Given that *Flaviviridae* includes numerous human and animal pathogens of high public health importance, the characterization of novel, divergent members like HSTV is critical for understanding viral diversity, evolution, and spillover potential. In particular, elucidating the structure and function of HSTV proteins is essential to assess its pathogenic mechanisms, identify diagnostic targets, and evaluate its relationship to known pathogenic viruses.

The structural region of the HSTV polyprotein resists conventional domain annotation based on sequence homology. Structural proteins of viruses are frequently poorly conserved even within the same family, a feature often attributed to adaptive evolution for efficient host cell entry and immune evasion [36]. Consistent with this trend, we were unable to identify any sequence motifs or domains in HSTV polyprotein that are characteristic of known structural proteins from the *Flaviviridae* family. To circumvent this limitation, we employed membrane topology prediction to delineate potential structural units within the N-terminal region (residues 1–1250) of the HSTV polyprotein. This approach identified four distinct extramembrane and cytoplasmic segments. However, subsequent three-dimensional modeling using AlphaFold3 yielded a high-confidence structure only for the N-terminal segment (SP1, residues 1–173). This outcome may reflect two non-mutually exclusive factors. First, the limited availability of experimentally solved structures of *Flaviviridae* structural proteins, particularly from divergent or recently discovered members, may constrain AlphaFold3′s ability to generate accurate models for highly divergent sequences. Second, the HSTV polyprotein may harbor novel protein domains that have not been previously described in the *Flaviviridae* family, thereby lacking detectable homology to known protein folds.

Phylogenetic analyses place HSTV, BoTV4, and Trinbago virus within the same emerging clade of LGF, distinct from *Pestivirus* genus. However, to date, no functional or structural annotation exists for the structural proteins of BoTV4 or Trinbago virus, their polyproteins remain entirely uncharacterized at the protein level. Notably, prior studies have shown that the HSTV nonstructural proteins exhibit structural similarity to those of viruses in the *Orthoflavivirus* genus [8]. In contrast, our findings reveal that the identified HSTV SP1 displays the highest structural resemblance to Npro and Erns, which are unique to the *Pestivirus* genus. Given the high sequence conservation across the N-terminal regions of HSTV, BoTV4, and Trinbago virus polyproteins, it is plausible that they encode structurally and functionally homologous N-terminal proteins.

Pestiviral Erns consists of an N-terminal ectodomain that folds into an RNase catalytic domain and a C-terminal transmembrane anchor that tethers the protein to the host cell membrane during infection [29]. The C-terminal region includes a transmembrane helix followed by a 23 a.a. amphipathic α-helix, which plays a critical role in membrane association [37]. Strikingly, our structural model of monomeric HSTV SP1 exhibits a highly analogous architecture: an N-terminal structured ectodomain (residues 1–173), a transmembrane part, and a downstream amphipathic α-helix. Also, Erns functions as a homodimer stabilized by an intermolecular disulfide bond between Cys171 residues of adjacent protomers [29]. In contrast, our attempts to model a dimeric form of HSTV SP1 were inconclusive. The resulting models varied substantially depending on the construct length and dimerization protocol, and none displayed a disulfide linkage, suggesting that SP1 may not form dimers via the same mechanism as Erns. Furthermore, the putative catalytic site of SP1 shows strong structural conservation with the active site of Erns [38]. Despite this similarity, the RNase activity of HSTV SP1 remains to be experimentally verified and was not assessed in the present study.

Notably, HSTV SP1 is encoded at the very N-terminus of the HSTV polyprotein, whereas Erns is not positioned at the leading end of the polyprotein in any known pestivirus [36]. This difference in genomic context, combined with the absence of dimerization and the unconfirmed enzymatic activity, indicates that while SP1 shares significant structural features with Erns, it may represent a functionally divergent or evolutionarily distinct variant.

HSTV SP1 also exhibits limited structural similarity to Npro, the N-terminal pestiviral autoprotease, which is encoded at the very beginning of the viral polyprotein, similar to HSTV SP1 [35,39]. Structurally, Npro adopts a two-domain “molluscan shell” fold, in which the N-terminal domain harbors the catalytic activity and the C-terminal domain mediates substrate binding. The Npro catalytic site is formed by a His49–Cys69 dyad, which is essential for its autoproteolytic function [40,41]. Similarly, HSTV SP1 displays a two-domain architecture that folds into a molluscan shell-like conformation. However, its putative active site shows virtually no structural or sequence conservation with the canonical Npro catalytic dyad. Moreover, functional assays under zinc-supplemented conditions failed to detect any proteolytic activity of recombinant HSTV SP1; specifically, no autocleavage of the chimeric fusion protein was observed, as expected [42].

Moreover, SAXS-based structural analysis of HSTV SP1 did not support significant similarity to the crystallographic structures of putative pestiviral homologs Erns and Npro. Quantitative comparison yielded reduced χ^2^ values of approximately 10 for Erns and as high as 24 for Npro, indicating substantial structural divergence between HSTV SP1 and these canonical pestiviral proteins (Appendix A). At the same time, the theoretical scattering curve generated from the RoseTTAFold-predicted HSTV SP1 structural model exhibited a good fit to the experimental SAXS data, with a reduced χ^2^ value of 1.5. A χ^2^ value of 1.5 indicates good agreement between the predicted structural model and the experimental SAXS data, confirming that the RoseTTAFold model accurately represents the tertiary architecture of HSTV SP1 in solution (Appendix A). The slight deviation from the ideal χ^2^ = 1 may be attributed to the presence of an N-terminal hexahistidine tag and a short intrinsically disordered C-terminal part in the recombinant HSTV SP1, both of which contribute to the scattering profile and can elevate the χ^2^ value despite an otherwise well-folded core structure. Nevertheless, it should be noted that the recombinant HSTV SP1 construct used in this study may not correspond to a naturally occurring viral protein. Membrane topology prediction provides only an approximate guide to domain organization within the viral polyprotein, and the exact boundaries of structural proteins in HSTV remain to be confirmed by direct experimental evidence.

In summary, while HSTV SP1 shares certain topological and functional features with pestiviral proteins Erns and Npro, it exhibits significant structural divergence from their canonical crystallographic folds. The agreement between experimental SAXS data and the AI-predicted model supports the conclusion that HSTV SP1 probably adopts a distinct spatial architecture in solution, one that is not captured by existing pestiviral structures but is reliably represented by modern AI-based prediction. This highlights the novelty of HSTV SP1 and the need to integrate computational modeling with experimental validation for the structural characterization of novel viral proteins lacking close structural homologs.

While this study provides the first structural characterization of an N-terminal protein from the emerging LGF clade, several key questions remain unresolved. Most notably, the membrane topology of SP1, and consequently its subcellular localization and functional context, cannot be deduced from the soluble SP1 construct used here. In pestiviruses, the topological dichotomy between cytoplasmic Npro and luminal Erns has profound implications for protein function and virus assembly. Moreover, functional validation, particularly regarding potential RNase or protease activity, receptor binding, or capsid-like RNA packaging, awaits the development of reverse genetics systems for HSTV and related LGF viruses. This work thus represents a foundational step, enabling future studies in virology and structural biology.

## 5. Conclusions

In this study, we report the first identification, recombinantly produced, and structurally characterized of SP1, a putative structural protein encoded at the N-terminus of the HSTV polyprotein. Despite the absence of significant sequence homology to known *Flaviviridae* proteins, integrative analysis revealed that HSTV SP1 adopts a compact, predominantly globular fold in solution, as confirmed by AI modeling, SEC, DLS, and SAXS. Structural comparisons indicate distant similarity to pestiviral proteins Erns and Npro, with substantial divergence in both domain architecture and functional motifs. Our findings indicate that HSTV SP1 adopts a fold not previously observed among characterized members of the *Flaviviridae* family, although its exact biological role and processing in the context of the native viral polyprotein remain to be confirmed. The structural model of HSTV SP1 not only advances understanding of HSTV biology but also offers a critical reference for annotation and functional prediction of uncharacterized N-terminal proteins in closely related LGF members, including BoTV4 and Trinbago virus.

This work highlights the importance of combining AI-driven prediction with biophysical validation to explore the structural landscape of emerging viral proteins, particularly those lacking close homologs in existing databases.

## Figures and Tables

**Figure 1 biomolecules-15-01690-f001:**
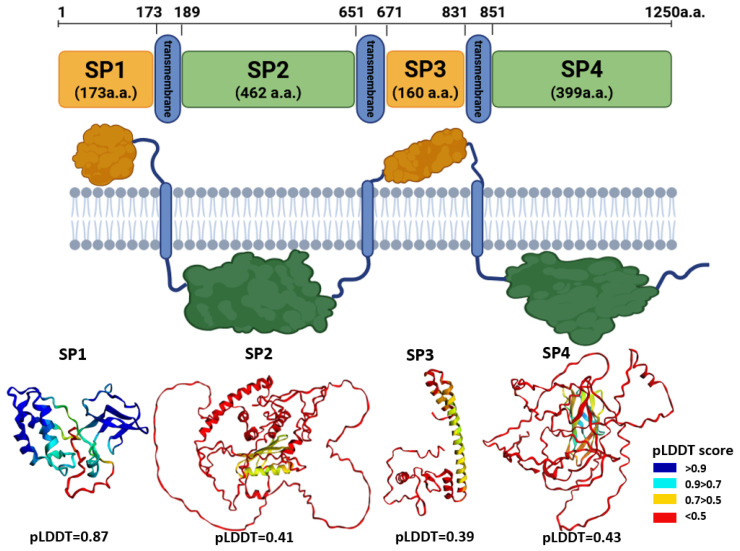
Predicted membrane topology of the putative structural region (residues 1–1200) of the HSTV polyprotein and AlphaFold3-predicted structural models of its extramembrane (orange) and cytoplasmic (green) parts. Transmembrane helices are shown as blue cylinders. Structural models are colored from blue to red accordingly pLDDT confidence score.

**Figure 2 biomolecules-15-01690-f002:**
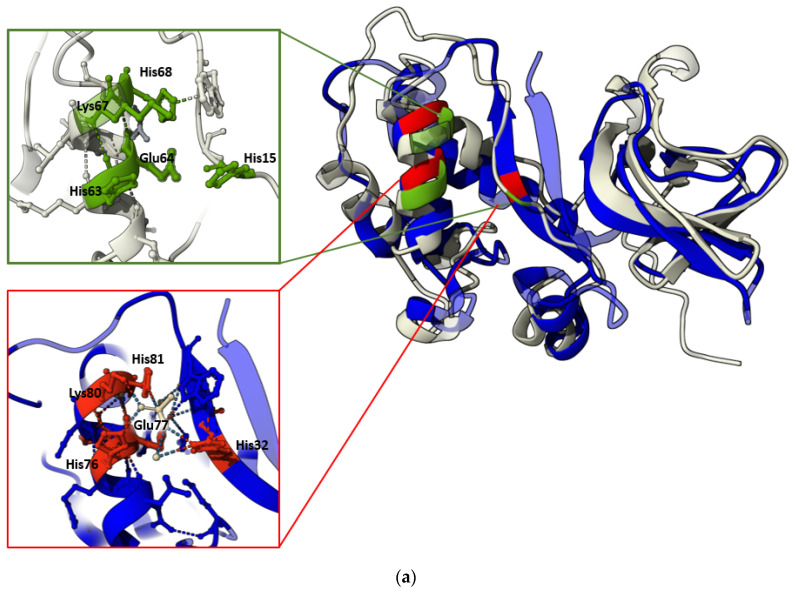
Comparison of primary and tertiary structures of HSTV SP1 with pestiviral Erns. (**a**) Structural superposition of the HSTV SP1 model (gray) and the BVDV-1 Erns (PDB ID: 4DVK, blue). Catalytic residues of BVDV-1 Erns are shown in red (red box). Structurally equivalent residues in HSTV SP1 are highlighted in green (green box). (**b**) Sequence conservation analysis of HSTV SP1 and pestivirus Erns: red—identical in all sequences residues; yellow—similar residues; white—unaligned residues. Catalytic residues are marked in green box. (**c**) Predicted disulfide bond network in the HSTV SP1 monomer. Cysteine pairs forming disulfide bridges (Cys28–Cys70, Cys57–Cys58, Cys99–Cys125, Cys95–Cys142) are indicated.

**Figure 3 biomolecules-15-01690-f003:**
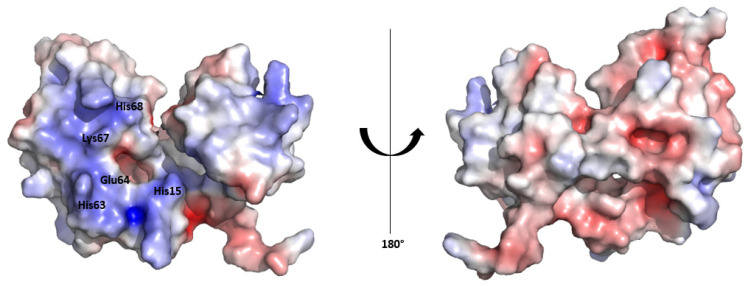
Electrostatic surface potential of the HSTV SP1. The positive surface potential is colored blue, and the negative surface potential is colored red. Putative HSTV SP1 catalytic residues are shown.

**Figure 4 biomolecules-15-01690-f004:**
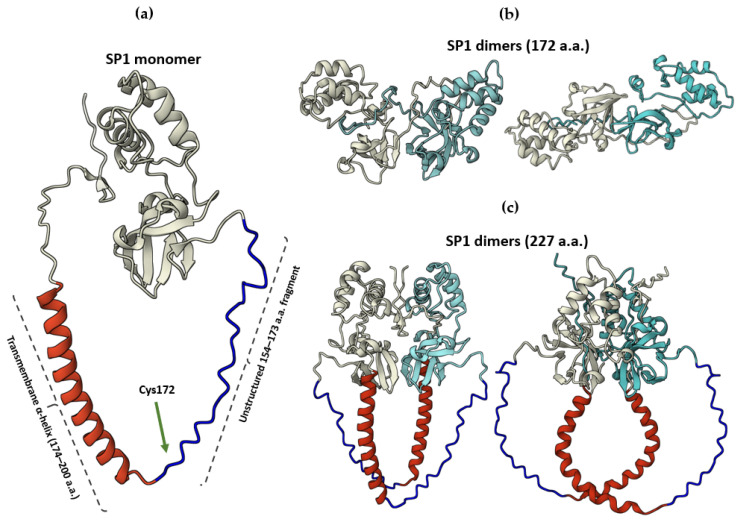
Predicted spatial structures of HSTV SP1 in monomeric and dimeric forms. (**a**) Monomeric HSTV SP1 model (227 a.a.) showing the structured ectodomain (gray; 153 a.a.), the unstructured region (dark blue; 20 a.a.), and the C-terminal transmembrane amphipathic α-helix (red; 26 a.a.). (**b**) Different dimeric HSTV SP1 models comprising only the ectodomain (172 a.a. per protomer; protomers colored gray and light blue). (**c**) Different dimeric HSTV SP1 models (227 a.a. per protomer) showing the structured ectodomain (gray and light blue; 153 a.a.), the unstructured region (dark blue; 20 a.a.), and the C-terminal transmembrane amphipathic α-helix (red; 26 a.a.).

**Figure 5 biomolecules-15-01690-f005:**
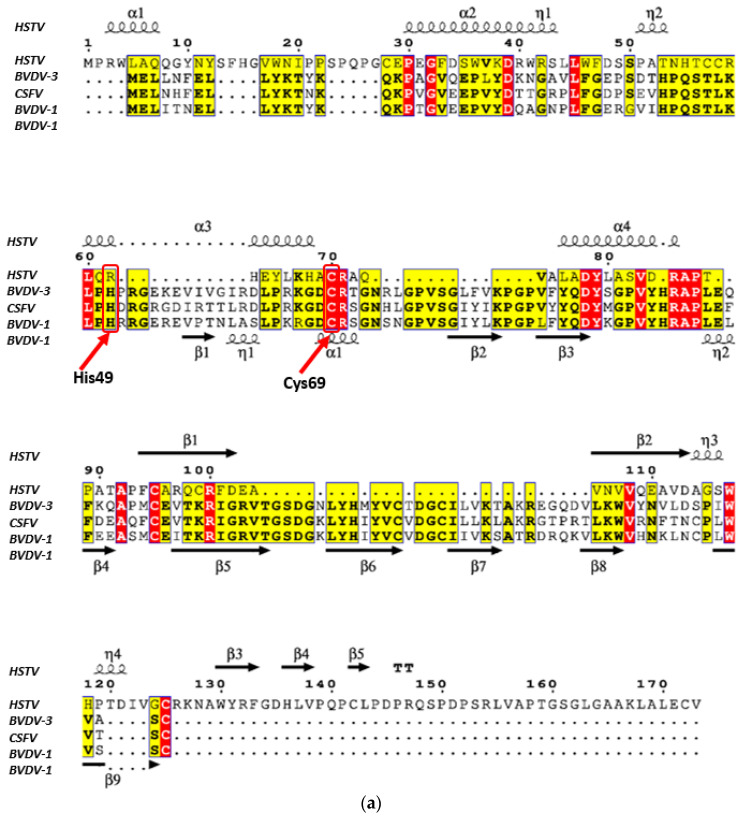
Comparison of primary and tertiary structures of HSTV SP1 with pestiviral Npro. (**a**) Sequence conservation analysis of HSTV SP1 and pestiviral Npro: red—identical in all sequences residues; yellow—similar residues; white—unaligned residues. Catalytic residues His49 and Cys69 are shown. (**b**) Structural models of BVDV-1 Npro (PDB ID: 4H9K, orange) with catalytic residues His49 and Cys69 (purple) and HSTV SP1 (gray) with structural position of Cys70 (green).

**Figure 6 biomolecules-15-01690-f006:**
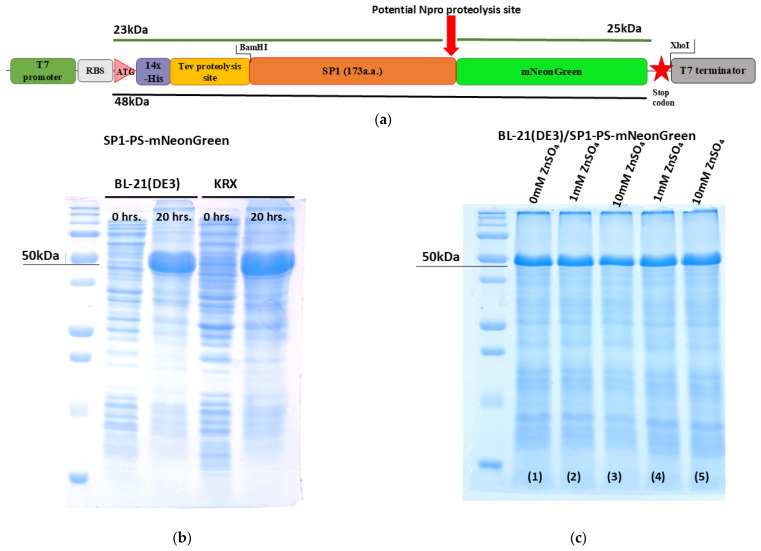
Schematic representation of the chimeric recombinant 14×His-TEV-SP1-PS-mNeonGreen protein designed to assess potential autoproteolytic activity of HSTV SP1. (**a**) Linear map of the T7-14×His-TEV-SP1-PS-mNeonGreen expression cassette, designed to produce the chimeric recombinant 14×His-TEV-SP1-PS-mNeonGreen protein. The red star indicates the stop codon. (**b**) SDS-PAGE of total protein lysates from BL21(DE3)/SP1-PS-mNeonGreen and KRX/SP1-PS-mNeonGreen producer strains following induced expression. Lane 0: uninduced control. Lane 20: cells harvested after 20 h. of induction. (**c**) SDS-PAGE of purified fractions from total protein lysates BL21(DE3)/SP1-PS-mNeonGreen cultured under varying zinc conditions. Lanes: 1—standard conditions cultured strain without ZnSO_4_; 2—+1 mM ZnSO_4_; 3—+10 mM ZnSO_4_; 4—purified protein with 1 mM ZnSO_4_; 5—purified protein with 10 mM ZnSO_4_.

**Figure 7 biomolecules-15-01690-f007:**
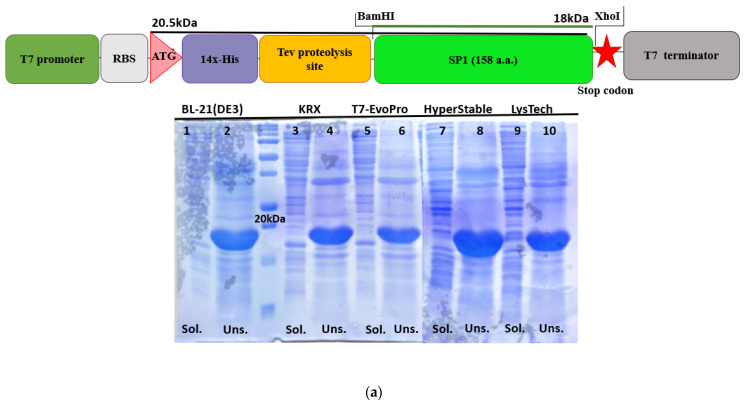
Design of expression cassettes and initial expression screening of recombinant chimeric HSTV SP1. (**a**) Linear map of the 14×His-TEV-SP1 expression cassette and SDS-PAGE of total protein lysates from BL21(DE3)/14×His-Tev-SP1, KRX/14×His-Tev-SP1, T7-EvoPro/14×His-Tev-SP1, HyperStable/14×His-Tev-SP1, and LysTech/14×His-Tev-SP1 producer strains. Odd-numbered lanes correspond to the soluble (sol.) protein fraction after expression, while even-numbered lanes represent the unsoluble (uns.) inclusion body fraction. (**b**) Linear map of the pEASY-HSTV-SP1 expression cassette and SDS-PAGE of total protein lysates from BL21(DE3)/pEASY-HSTV-SP1, KRX/pEASY-HSTV-SP1, and HyperStable/pEASY-HSTV-SP1 producer strains. Lanes 1–5 correspond to the expression temperature: 1—37 °C, 2—30 °C, 3—25 °C, 4—20 °C, 5—16 °C. (**c**) Linear map of the pET-HSTV-SP1 expression cassette and SDS-PAGE of total protein lysates from BL21(DE3)/pET-HSTV-SP1 and KRX/pET-HSTV-SP1 producer strains. Lane numbers correspond to the cultivation time after induction. The red star indicates the stop codon.

**Figure 8 biomolecules-15-01690-f008:**
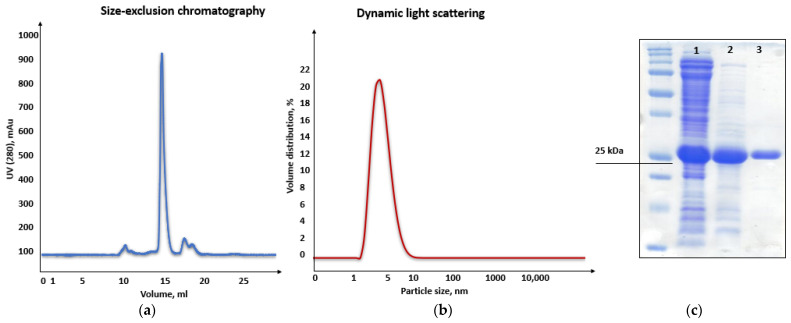
Purification and biophysical characterization of recombinant His-tagged HSTV SP1. (**a**) Size-exclusion chromatography (SEC) elution profile; (**b**) Dynamic light scattering (DLS) analysis of the final HSTV SP1, showing a monodisperse population with a hydrodynamic diameter of 4.4 ± 0.1 nm. (**c**) SDS-PAGE of the purified SP1 fractions: lane 1—cell lysate before purification, lane 2—fraction of proteins after ion-exchange chromatography, lane 3—fraction of SP1 protein after SEC.

**Figure 9 biomolecules-15-01690-f009:**
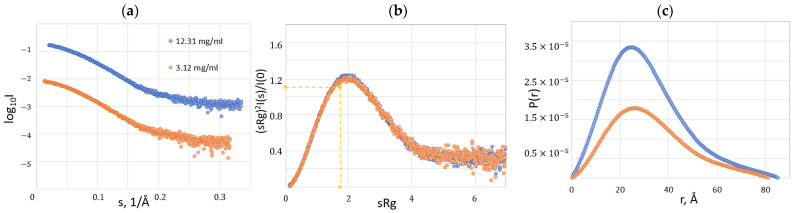
SAXS analysis of the HSTV SP1 at concentrations of 12.31 mg/mL (blue) and 3.12 mg/mL (orange). (**a**) Experimental SAXS profiles of HSTV SP1. (**b**) Kratky plot of HSTV SP1. The point marked by the intersection of the yellow dashed lines corresponds to the value expected for a fully folded (globular) protein. (**c**) Distance distribution function—P(r) of HSTV SP1. All curves are normalized to protein concentration.

**Figure 10 biomolecules-15-01690-f010:**
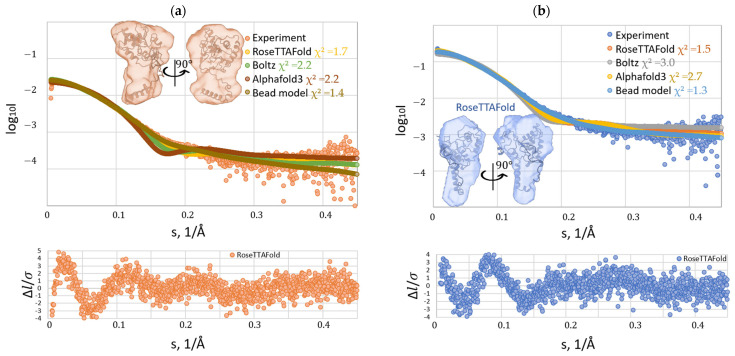
Comparison of experimental SAXS curves (dots) with Crysol-computed scattering profiles from AI-predicted models (RoseTTAFold, Boltz-2, AlphaFold3) and the DAMMIN bead model for HSTV SP1: (**a**) at concentrations of 3.12 mg/mL (orange) and (**b**) at concentrations of 12.31 mg/mL (blue). Theoretical fits are colored as labeled, with corresponding χ^2^ values indicating the goodness-of-fit. Structural models (ribbon representation) are shown above for reference. Each structural model is superimposed onto the corresponding experimental SAXS-derived envelope (transparent orange and blue envelopes). The lower panel displays the normalized residuals (ΔI/σ) between RoseTTAFold-predicted HSTV SP1 model (chosen as the best fit) and experimental data as a function of the scattering vector s.

**Table 1 biomolecules-15-01690-t001:** Structural comparison of HSTV SP1 with the Erns of BVDV-1 based on pairwise structural alignment.

PDB ID *	TM-Score	RMSD, Å	Length, a.a.	Aligned Residues, a.a.
4DVK	0.69	3.51	158	120
4DW5	0.64	3.29	161	120
4DW3	0.63	3.61	154	111
4DWA	0.63	3.49	157	115
4DWC	0.62	3.49	156	113
4DW7	0.62	3.55	161	113
4DVL	0.61	3.57	165	112
4DVN	0.61	3.61	154	110
4DW4	0.61	3.54	157	113

* All currently available pestiviruses Erns structures deposited in the PDB originate from a single study ([29]).

**Table 2 biomolecules-15-01690-t002:** Overall structural parameters of the HSTV SP1 obtained from SAXS data.

Concentration, mg/mL	$R_{g}$, nm	$D_{\max}$, nm	$\mathrm{Excluded}\;V\;*,\;A^{3}$	MW(Exp) *, kDa	MW(Theory) *, kDa
3.12	2.5 ± 0.1	8.0	$43\times 10^{3}$	33.1 [CI: 31.3, 35.0]	27
12.31	2.4 ± 0.1	8.4	$48\times 10^{3}$	28.9 [CI: 27.9, 30.6]	

* Excluded V is the particle volume, MW (exp) is the experimental molecular weight, and MW (theory) is the theoretical molecular weight calculated according to the HSTV SP1 a.a. sequences.

**Table 3 biomolecules-15-01690-t003:** Comparison of experimental HSTV SP1 SAXS profiles with theoretical scattering curves calculated from structural models generated by AlphaFold3, Boltz-2, and RoseTTAFold.

SP1 Model	$R_{g}$, nm	$D_{\max}$(Envelope), nm	$\mathrm{Volume},\;A^{3}$	Concentration: χ^2^
AlphaFold3	Shell *R*_g_: 2.6 Envelope *R*_g_: 2.6	11.1	Excluded Volume: $31\times 10^{3}$ Envelope Volume: $44\times 10^{3}$	12.31 mг/mл: 2.7
				3.12 mг/mл: 2.2
Boltz-2	Shell *R*_g_: 2.4 Envelope *R*_g_: 1.9	6.0	Excluded Volume: $31\times 10^{3}$ Envelope Volume: $36\times 10^{3}$	12.31 mг/mл: 3.0 3.12 mг/mл: 2.2
RoseTTAFold	Shell *R*_g_: 2.7 Envelope *R*_g_: 2.4	8.7	Excluded Volume: $31\times 10^{3}$ Envelope Volume: $46\times 10^{3}$	12.31 mг/mл: 1.5 3.12 mг/mл: 1.7

## Data Availability

The original contributions presented in this study are included in the article/Appendix A; further inquiries can be directed to the corresponding author.

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
