# Peer review of "First Identification, Recombinant Production, and Structural Characterization of a Putative Structural Protein from the Haseki Tick Virus Polyprotein"

_biomolecules, 2025, doi:10.3390/biom15121690_

Round 1
Reviewer 1 Report
Comments and Suggestions for Authors
There is a structural study of newly discovered viral proteins including protein structure prediction, following by experimental validation including diverse methods, such as expression of putative proteins (SP1, HSTV structural protein), SAXS, comparison of SAXS data with homologous proteins. Proposed workflow is reasonable and could be used for characterization of highly divergent protein families. Also these findings are valuable in the virus research field.
The main experimental data were obtained using SAXS. The described SAXS workflow is reasonable, data quality is good.
Still I would suggest some correction of the text and graphical representations, that, I believe, could be improved significantly by simple means.
I also recollect that it is one application from ATSAS package, DARA, which probably could be useful if authors will continue with SAXS characterization: http://dara.embl-hamburg.de/ ;(https://academic.oup.com/bioinformatics/article/32/4/616/1743953?login=true). I do not mean that this server should be cited or mentioned in this manuscript, but it is a software that performs the search in PDB by SAXS derived low resolution shape, which could be interesting in further development of this approach.
Lines 19-20
Not clear what is "the structural part of the HSTV polyprotein."? I would add a few words of explanation to non-viral readers in Introduction (I mean that polypeptide consists from many ORFs, N-terminal is coding structural proteins followed by RNA polymerase etc.).
Lines 212-216
Would be interesting to compare domain structure of Bole tick virus polyprotein with HSTV models. Are they similar or differ? I do not know, whether it is known/predicted/proved experimentally, but in case there is something, would be good to put few words here or into Introduction. Also Fig. 1S maybe is not so necessary. Instead, would not it be more informative to insert in Fig. S1 the domain structures of Bole tick virus or, maybe some other related viruses?
Fig. 1
Thin lines from domain structure to membrane location scheme could be removed. The transmembrane scheme below also could be omitted as superficial. Also unused is the green line above the domain scheme.
Anyway, authors can try to make a figure much more concise following an example from their previous publication [ref.8, Fig.8], which represents a fine and clear scheme of the position of structural proteins within polypeptide and their cellular location. I would also suggest to add a predicted location of transmembrane helices along the polypeptide map, leaving Fig. S2 as a raw data or delete it. Also, why the right boundary in the domain scheme is uncertain (~1200 aa), if in Table S2 stated 1250 aa? I would suggest to add maximum info into this scheme with exact boundaries, one can also add domain lengths as well (within boxes). Also would be fine to include into this scheme all modeled domains with exact boundaries: 1-172 (with subdomains?), 1-227 (with marked subdomains, TM and linker) and other SP.
2.1.2. SP1 tertiary structure model
After I finished this chapter, I think it is better to change the header into "SP1 tertiary structure models" having in mind that two constructs are described here: 1-173 and 1-227 (1-173 with adjacent TM at C-terminus?). I also think that it should be mentioned in this chapter introduction.
Lines 232-233
Authors can be more explicit with these domains, one can put here exact domains lengths and boundaries instead of pLDDT scores, which could be inserted into Fig. 1 along the models representations.
Fig. 2b
In the legend are not mentioned green numbers and other marks (gray stars) under alignment. Also would be useful to mark with any sign predicted catalytic residues. In overall, alignment could be formatted more carefully and with better resolution, maybe. Red frames are too thick and some residues are hidden. Also unclear is the meaning of "80% aligned residues" - 4 res. coincide from 5 compared? If there are 3 sequences in the alignment, then 80% is hard to calculate. Maybe one can replace it with "similar (or equivalent) residues"?
Lines 297-301
"..Inner and outer surface of protein..." "...The inner surface, which lines the concave fold..." ‒looks strange and confusing.
Maybe, it would be better to use the protein surfaces description from [29], convex face and concave faces of Erns protein?
Line 305-306
"The AlphaFold3-predicted model of the N-terminal 227 a.a. of the HSTV polyprotein..." This model appeared here somehow suddenly (should it be marked on domain structure diagram?) in addition to the previously described putative N-terminal RNase-like domain (SP1?). In Table S2 there is no such domain boundaries like 227. Maybe authors can add more explanations on this point. It looks like there are two models of SP1: 1-173 and 1-227. Then which one is represented in Fig. 1?
Line 313
"...homodimers analogous to Erns..." Are there crystallographic dimers between structures in [29]? If there are dimeric Erns, it would be interesting to compare in Fig. 4 crystallographic and predicted dimers.
Fig. 4
Formatting of panels looks strange. I would suggest to combine panels in some graphical software and insert a complete Figure as one image. Also would be interesting to compare dimers of both constructs (1-172 and 1-227) with crystallographic dimers of Erns (if present) by spatial superposition, maybe in Supplementary or in additional panel.
Fig. 4a
The view makes an impression that TM helix is bound to N-term of the protein, would be fine to rotate slightly this view. Also one can add the boundaries of TM helix like those for linker.
Fig. 4b
If there is two rotational views of the same dimer (it is not clear), then the label "SP1 dimers" should be corrected to "dimer" and rotation axis should be inserted. If there are two variants of dimer, it should be mentioned somewhere, and corrected in the legend, where "dimeric model" is in singular.
Fig. 4c
227 aa is the length of a monomer in the dimer shown here? The label on this panel is "SP1 dimers". But it looks more like there are two views of one dimer after rotation. If so, should be corrected.
Table 2
The Table is the fragment of the Table S3, I would suggest to remove it from text.
Line 338
"...SP1 comprises two domains that together adopt a conformation resembling a molluscan shell."
Which dimer is discussed here? AlphaFold or from crystal structure? Please, specify.
Lines 339-340
"Furthermore, the C-terminal domain of SP1 exhibits a secondary structure arrangement similar to that of the CSFV Npro."
From the alignment in Fig. 5 it does not look like secondary structures are similar, except beta strands at ~90-100 and ~110 aa, whereas it is obviously similar with Erns (Fig.2). This statement is unclear, and needs some more illustration or comment.
Line 343
"...Cys69 and a zinc ion, forms a hydrogen-bonded catalytic triad" In Discussion (Line 574) it is a catalytic dyad.
Line 350
"Catalytically residues His49 and Cys69 are shown." Please, correct typo. Also a green number should be explained in legend.
Fig. 5a
Same problems as in Fig. 2b: if 3 from 4 coincide, then it can be 75% aligned residues, for 80% 5 sequences should be compared. Also in the legend is not explained the meaning of bold text, black stars and green numbers. Maybe it would be more informative to make bold or colored the identical in all sequences residues, yellow - similar or equivalent residues (also in all 4) and white - not conserved ones? Or color coinciding/equivalent residues in all viruses but HSTV. Such coloring can emphasize the differences.
Fig. 5b
Overlay of active centers is not clear. Purple residues are not explained in the legend. Maybe better to make two separate views in the same orientation and make a structural superposition of both proteins thus demonstrating the similarity of fold (like in Fig. 2)? From the first glance views presented here can hardly support the high similarity.
Line 365
"...putative proteolytic cleavage site..." Would be fine to add more comment on this site, how variable it is? Maybe cleavage failed since there could be some variations in such sites?
Lines 388-389
"This finding is consistent with our earlier structural analyses, which revealed no significant similarity between the Npro and the SP1 model."
Albeit the result is negative, and there is no positive control for this assay (I appreciate, that it needs significant efforts to make such experiment, and I believe, it would be too much for this study). It is in line with low structural similarity, especially since the given views are indeed not convincing. But earlier, in Lines 329-330 it was claimed: "...striking topological resemblance in their domain organization (Table 2, Table S3)." Maybe authors can soften one of these statements?
Fig. 6
"Lanes: 1 – standard conditions". Maybe, there was no Zn added? Authors could make gel more understandable by providing description on top of gels. Thus, here it would be enough to add hrs to the line with induction times in “b” and Zn concentrations in “c”. Also thin line at 50 kDa probably shifted in final image?
Lines 397-398
"Also, to facilitate successful structural studies, the intrinsically disordered C-terminal region was removed from the HSTV SP1-encoding sequence (Fig. 7)."
The regions at termini usually are poorly modeled, since they usually lack the defined secondary structure. But it does not mean that such regions are disordered. Here I would consider the overlay of several crystal structures and look which parts of the proteins are not resolved. Indeed, very often terminal parts are absent in structures. Terminal loops are also often glued to the protein surface. Here (or in Fig. 7a), I think, it will be informative to define explicitly the boundaries of the construct.
Fig. 7
For clarity would be fine to mark soluble/unsoluble (with letters?) on the top or bottom of gels. The same for temperature in Fig. 7b and time in 7c. There is enough place to write all this information on the panels.
Line 429
Maybe, put IMAC into abbreviation list?
Fig. 8a
Peak numbers in the chromatogram are not clear. Which peak corresponds to the Sp1 protein? Actually more information is present in the legend of Fig. S9. Therefore, maybe it is worth considering to remove these unexplained gels and chromatograms from the main text and leave the Supplementary figures with more careful formatting? In the main text I would suggest to leave only gel (with proper description) from Fig. S10.
Fig. 8b
Gel needs more explanations. Do the numbers on top correspond to the fraction numbers? Which chromatography is presented here? Which spots correspond to the product? Actually later I found explanations in Fig. S9. Here the same recommendation could be the same: combine chromatogram with gel and put peak number on top of the gel or make projection from peaks on chromatogram onto gel directly. It will make a purification story more clear.
Fig. 8e
It is not even mentioned in the legend, and not clear where it is in the Figure, probably the small gel on DLS? Later I found it in Fig. S10 (see comment on Fig. 8a).
Line 459
Dmax measured by SAXS was 8-8.4 mn, whereas DLS gives "monodisperse population with a hydrodynamic diameter of 4.4 ± 0.1 nm" (Fig. 8 legend). Could it be possible, that it was a radius measured by DLS?
By the way, and it is not mentioned in the Methods section, which program was used for MW estimation by Bayesian method (would be fine to put into Methods section)? Actually there are many methods: SAXSMoW, PRIMUS, where many methods are combined. Actually often they give different results, usually in the case of good compact protein most of results are compatible. Also DAMMIF/ DAMMIN could be used for MW estimations and often give quite good results.
Line 462 and 464
Which "inaccuracies in theoretical mass calculations" could be there? Why not to calculate actual MW from the actual construct sequence? The theoretical MW is somehow quite certain and should include all expression tags, that are not cleaved. But the visible increase in SAXS MW could be a result of many factors, such as hydration and shape (elongated, disordered tails, cavities etc.), main source of uncertainty is usually protein aggregation (which is not the case here) and concentration errors (since I(0) depends on concentration, and MW is estimated from I(0)). Actually, 6His-tag should be included in the MW calculation especially for such a small protein.
Table 3
I think that it would be useful to put here (or in Methods section) the methods/software used for estimation of SAXS parameters. For example, Rg is estimated from Guinier approximation or from GNOM parametrization, Volume and Dmax too could be estimated by several methods.
Fig. 9
Interestingly, did the authors normalized both curves by concentration? In this case one can produce a merged scattering curve with the initial part from the low concentration, and high s data from high concentration. Also, in this case p(r) functions were also normalized, which could better demonstrate data quality, which seems to be good in this case. But it is not necessary, maybe, for this study.
Line 488
"...theoretical scattering profiles..." - it would fine to add here of which ones? AlphaFold models?
Fig. 10
One can improve the Figure by labeling predicted models as well as DAMMIF models derived from both experimental curves directly on panels. By the way, from inspection of this Fig. I get an impression, that αFold model fits best into DAMMIF envelops derived from both experimental scattering curves.
Also, I think, that the best way to understand the goodness of fit is the CRYSOL curves calculated from atomic models overlaid with experimental SAXS curves. One can also put χ2 values along with dummy atom DAMMIF models and (maybe) graphs of CRYSOL curves directly in the Figure.
The first impression looking on the Fig. 10 is that there are presented 6 different dummy-atom models. Actually, I would expect that there only 2 of them derived from both experimental curves. I would suggest to put DAMMIF models into the same scale and, if possible, to keep the same orientation. Current views are somehow chaotic.
Line 508
"...DAMMIF and dummy-atom modeling..." DAMMIF is the dummy-atom modeling software.
Line 515
"...guiding future functional and X-ray crystallography studies." Not quite clear how SAXS data can guide crystallography.
Lines 589-590
"RoseTTAFold model accurately represents the tertiary architecture of HSTV SP1 in solution"
Here would be fine to see a superposition of both models (maybe in Supplement?) to see differences between them.
Lines 609-611
"In this study, we report the first identified, recombinantly produced, and structurally characterized of SP1,..."
Fig. S1
Is this figure so much necessary in the text?
Fig. S2
Would be fine to add x-axis definition to the legend, whether there are amino acids or nucleotides? And whether it is a complete polypeptide or complete virus? And how domain structure in Fig.1 is related to this scheme?
Table S2
Would be fine to add here also domains SP1-SP4. Otherwise, authors can consider the placement of more precise domains and TM regions boundaries in the domain scheme of Fig.1 and this will make this Table unnecessary.
Table S4, S5
Not clear the meaning of +/-, it should be explained.
Fig. S6
It would be less confusing, if authors mark soluble/unsoluble fraction on the top of gels (something like "s", "u" or other), since in the second gel odd and even lines are shifted.
Fig. S13
I would suggest to put here CRYSOL calculated scattering curves from models on the experimental data of both concentrations (but maybe for CRYSOL one have to reduce concentration?). Since χ2 values are much better illustrated by direct curves comparison. Actually lowest χ2 one can see for the worst measurement with large scattering of experimental points, since in this case thin theoretical curve with be completely within the experimental data. From the other hand, the thin experimental curves with very good statistics will coincide with theoretical calculated curves in less points.
Again, the Figure would gain if protein names and PDB ID were put as labels on panels.
Also legend could be re-formulated like this, for example:
Low-resolution ab initio molecular envelopes of HSTV SP1 in solution reconstructed from SAXS using DAMMIF (transparent purple, 3.12 mg/ml data) superimposed with atomic models (or better - with crystal structures of ... viral proteins: a, b, c) superimposed on the corresponding SAXS-derived consensus envelope:
Ref.12
Please, correct format.
Author Response
Thank you for reviewing our manuscript (biomolecules-3973405) entitled “First Identification, Recombinant Production, and Structural Characterization of a Putative Structural Protein from the Haseki Tick Virus Polyprotein” submitted for publication in Biomolecules.
We thank you for valuable suggestions that allowed us to make the manuscript more convincing and understandable. We accepted your suggestion and made corresponding change in the manuscript. We major revised our article. Below please find our detailed responses to your questions and comments. All modifications in the manuscript have been highlighted in red.

Reviewer 2 Report
Comments and Suggestions for Authors
The manuscript presents the first experimental characterisation of SP1, a putative structural protein domain from Haseki tick virus (HSTV), a recently detected, potentially zoonotic member of the Flaviviridae family. The authors use bioinformatic predictions, AI-based structure modelling (AlphaFold3, RoseTTAFold, Boltz-2), and experimental validation (recombinant expression, purification, and SAXS analysis). SP1 was the only predicted fragment successfully produced and structurally analysed, showing partial similarity to pestiviral Erns and Npro proteins but lacking their catalytic motifs. SAXS confirmed a compact, likely monomeric protein, supporting the structural predictions. The study is novel, combining AI modelling and SAXS to tackle the structure of a highly divergent viral protein. However, it overstates functional claims and contains methodological weaknesses that require a major revision before publication.
Major issues
- Please avoid the overstatement of novelty and function of SP1(Lines 24–32, 599–607, 609–621): The manuscript repeatedly concludes that SP1 is a “structurally novel protein within the Flaviviridae family” and implies it is a bona fide “structural protein” of the virion. Yet no virion context (particle incorporation, surface exposure, secretion, membrane anchoring in infected cells) is experimentally demonstrated. SAXS confirms a folded soluble domain, not its role in virion assembly.
- Lines 543–548: The manuscript suggests “a complex evolutionary history for HSTV, potentially involving recombination events or convergent evolution... features characteristic of different Flaviviridae members.” At present, this statement is speculative.
- Lines 255–318 and 326–347, 570–585: The results spend considerable space comparing SP1 to Erns and Npro. Later, in Discussion, you state that SP1 shows “features reminiscent of both Erns and Npro pestiviral proteins, yet these similarities are insufficient for definitive functional annotation.
- Lines 360–389: You conclude that “SP1 lacks detectable autoproteolytic activity”. The construct used is SP1 fused to mNeonGreen with a presumed cleavage site and tested in Zn²⁺-supplemented conditions. This needs clarification, for example:
(a) What exact sequence constitutes the presumed cleavage site (lines 362–366 refer to “including a putative proteolytic cleavage site” but do not define its position or motif).
(b) Whether the fusion orientation mimics the authentic polyprotein context (i.e. SP1 N-terminus / C-terminus relative to natural flanking regions in HSTV).
(c) The detection sensitivity: could low-level cleavage have been masked by co-migration on SDS–PAGE (~49 kDa band, lines 368–377)?
(d) Whether you also attempted in vitro incubation post-purification (rather than only in-cell expression lysates). In my opinion, without these clarifications, the negative result is hard to interpret - Lines 189–210, 448–498: The SAXS experiment is central to the manuscript’s claim that SP1 adopts a compact monomeric fold consistent with AI predictions. However, several methodological details that are standard in structural biology papers are either missing
(i) Exact sample buffer at the beamline (you mention 20 mM Tris-HCl, 150 mM NaCl, pH 8.0 in lines 196–198, but please confirm whether any additives, reducing agents, glycerol, etc., were present during data collection).
(ii) Temperature during data collection.
(iii) Whether frames were averaged after checking for radiation damage (20 × 1 s exposures are mentioned, lines 196–198).
(iv) The q-range actually used for Guinier fits and for P(r) estimation (you report q-range 0.07–4.5 nm⁻¹ in lines 194–195 but do not specify truncation criteria for Guinier).
(v) Definition of χ² used for model fitting (reduced χ²? weighted? which program?). - Lines 497–507 and Figure 10: You superimpose AlphaFold3, Boltz-2, and RoseTTAFold models into ab initio SAXS envelopes and then discuss “burial” of the His-tag in the Boltz-2 model to explain IMAC failure (lines 503–506). This is interesting, but currently speculative. You have not experimentally mapped the His-tag accessibility (e.g. limited proteolysis, anti-His western blot on native vs denatured protein).
Minor issues
- Lines 187–188 and 437–440: You report hydrodynamic diameter 4.4 ± 0.1 nm. Please also report the polydispersity index (PDI) / %mass in main peak if available, to support the claim of monodispersity.
- The manuscript presents excessive number of figures. The main text should include only the main findings. Remaining data could be included in supplementary materials.
Author Response

(The authors gave the same response as above.)

Reviewer 3 Report
Comments and Suggestions for Authors
The manuscript describes the production and structural characterization of a structural protein from HSTV. The paper is well written and has a logical flow. My minor comments are as follows:
- The abbreviation HSTV (Haseki tick virus) is defined multiple times throughout the text. It should be defined only once, upon first mention in the Introduction, and then consistently referred to as HSTV
- In the Figure 6C legend, it is stated that the panel shows SDS-PAGE of total cell lysates and purified fractions. However, both appear quite similar, making it difficult to distinguish between them. Please clarify which lanes correspond to total lysate and which to purified fractions. Additionally, specify which purification method was used, as this information is currently unclear.
- The reported lack of proteolytic activity in the presence of ZnSO₄ is not fully convincing without a positive control. Please clarify or discuss this point further.
- Figure 7 could be made clearer. For example, it is not explained what the 3C proteolysis site or the Xpress tag A brief explanatory sentence describing these elements would greatly improve comprehension.
- Approximately one-third of the recombinant protein consists of N-terminal tags and the cleavage site. How was this accounted for during structure modeling using SAXS? Please clarify.
- I am also wondering why the authors did not use the protease cleavage site to remove the tags, which would have yielded a more naturally occurring SP1 for the SAXS study.
Author Response
Thank you for reviewing our manuscript (biomolecules-3973405) entitled “First Identification, Recombinant Production, and Structural Characterization of a Putative Structural Protein from the Haseki Tick Virus Polyprotein” submitted for publication in Biomolecules.
We thank you for valuable suggestions that allowed us to make the manuscript more convincing and understandable. We accepted your suggestion and made corresponding change in the manuscript. We revised our article. Below please find our detailed responses to your questions and comments. All modifications in the manuscript have been highlighted in red.

Round 2
Reviewer 2 Report
Comments and Suggestions for Authors
The new version of the manuscript has incorporated all reviewer's comment in a clear manner. From my experience, the manuscript can be considered for publication in Biomolecules.
Author Response
We sincerely thank the reviewer for their thorough, insightful, and constructive feedback, which significantly improved the quality and rigor of our manuscript.